# The Associations between Snack Intake and Cariogenic Oral Microorganism Colonization in Young Children of a Low Socioeconomic Status

**DOI:** 10.3390/nu16081113

**Published:** 2024-04-10

**Authors:** Ahmed Alkadi, Naemah Alkhars, Samantha Manning, Hongzhe Xu, Michael Sohn, Jin Xiao, Ying Meng

**Affiliations:** 1Eastman Institute for Oral Health, University of Rochester Medical Center, Rochester, NY 14642, USA; ahj.alqadi@hotmail.com (A.A.);; 2Dental Department, King Fahad University Hospital, Imam Abdulrahman Bin Faisal University, Dammam 31441, Saudi Arabia; 3Department of General Dental Practice, College of Dentistry, Health Science Center, Kuwait University, P.O. Box 24923, Kuwait City 13110, Kuwait; naemah.alkhars@ku.edu.kw; 4Department of Biostatistics and Computational Biology, University of Rochester Medical Center, Rochester, NY 14642, USA; 5School of Nursing, University of Rochester, Rochester, NY 14642, USA

**Keywords:** snack intake, *Streptococcus mutans*, *Candida albicans*, oral health

## Abstract

Cariogenic microorganisms are crucial pathogens contributing to the development of early childhood caries. Snacks provide fermentable carbohydrates, altering oral pH levels and potentially affecting microorganism colonization. However, the relationship between snack intake and cariogenic microorganisms like *Candida* and *Streptococcus mutans* in young children is still unclear. This study aimed to assess this association in a prospective underserved birth cohort. Data from children aged 12 to 24 months, including oral microbial assays and snack intake information, were analyzed. Sweet and non-sweet indices based on the cariogenic potential of 15 snacks/drinks were created. Mixed-effects models were used to assess the associations between sweet and non-sweet indices and *S. mutans* and *Candida* carriage. Random forest identified predictive factors of microorganism carriage. Higher non-sweet index scores were linked to increased *S. mutans* carriage in plaques (OR = 1.67, *p* = 0.01), potentially strengthening with age. Higher sweet index scores at 12 months were associated with increased *Candida* carriage, reversing at 24 months. Both indices were top predictors of *S. mutans* and *Candida* carriage. These findings underscore the associations between snack intake and cariogenic microorganism carriage and highlight the importance of dietary factors in oral health management for underserved young children with limited access to dental care and healthy foods.

## 1. Introduction

Early childhood caries (ECC), which is recognized as the most prevalent chronic childhood disease, disproportionately affects up to 70% of preschool children from socioeconomically disadvantaged backgrounds in both developing and industrialized countries [1,2]. ECC can lead to long-term consequences to children’s health, such as pain during eating or drinking, difficulties in biting and chewing, and reduced appetite, potentially resulting in weight loss. Among the various factors contributing to early childhood caries, cariogenic microorganisms play a crucial role. *Streptococcus mutans* (*S. mutans*) is a well-established contributor due to its acidogenic and aciduric capabilities, as well as its role in forming the extracellular matrix, which is an essential component for dental plaque [3,4]. Another emerging cariogenic pathogen is *Candida albicans,* which has been found in high levels in children with early childhood caries [5,6]. *C. albicans* also exhibits acidogenic and aciduric properties [7,8]. Additionally, *C. albicans* has been found to be positively correlated with *S. mutans* carriage [9], possibly due to its ability to increase *S. mutans* in biofilms through a unique adhesive interaction mediated by extracellular polysaccharide formation. This interaction may lead to more severe caries when a person is co-infected with *S. mutans* [10,11,12]. 

A variety of dietary factors have also been linked to ECC. Research indicates that the sugar content, micronutrient composition, and frequency of eating can influence oral pH properties, buffering capacity, and enamel health [13]. Snack intake has emerged as a critical contributor to the elevated risk of ECC, given the significant increase in snacking frequency and energy intake among young children in recent decades [14,15,16,17]. Sweets and desserts continue to be the primary components of snacks in children [14], and the relationship between snack intake and ECC may be partially attributed to the ability of sugary snacks to modulate the presence and colonization of cariogenic oral microorganisms. These snacks with a high sugar content provide fermentable carbohydrates that stimulate bacterial growth and alter the pH levels in the oral environment, thereby promoting bacterial biofilm formation [18,19]. There, snack intake, particularly those rich in sugars, may play a role in fostering the growth of cariogenic microorganisms and contributing to the development of ECC.

Several studies have investigated the associations between sugary snacks and *S. mutans* carriage in young children, yielding mixed results [19,20,21,22,23,24,25]. However, there are gaps in our understanding, particularly regarding the relationships between snack intake and other cariogenic microorganisms, such as *Candida*. Additionally, while sugary snacks have been extensively studied, the individual effects of low- or non-sugary snacks have been largely overlooked. However, previous studies have indicated a link between eating frequency, which includes both sugary and non-sugary foods, and *S. mutans* carriage [20,25]. To address these knowledge gaps, this study aimed to assess the associations between snack intake, including snacks of high and low sugar content, and cariogenic microorganisms, including *S. mutans* and *Candida,* in a prospective cohort of young children from low socioeconomic backgrounds. 

## 2. Methods

### 2.1. Study Population

The current study is a subset of a parent prospective birth cohort study. The birth cohort was recruited from two university-affiliated clinics in upstate New York [26]. Pregnant women aged 18 years or older, eligible for New York State-supported medical insurance, and carrying a singleton fetus, were enrolled during the third trimester of pregnancy. Following birth, their full-term infants were also recruited for the birth cohort and assessed up to 24 months of age. Infants were excluded if they had a low birthweight (<2500 g), had Down syndrome, had orofacial deformity, or had received oral and/or systemic antifungal treatment before the baseline visit. A total of 160 infants were enrolled in the subsequent postnatal visits. For inclusion in the current study, infants had oral microbial data at 12, 18 or 24 months, as well as dietary assessment on snack intake at the same visits. A small number of records (*n* = 7 at 18 months and *n* = 10 at 24 months) without information on feeding method at 12 months were excluded from the current study (Figure 1). 

### 2.2. Oral Sample Collection and Quantification

Oral samples, including saliva and plaque, were collected at each visit. All care providers were informed not to brush infants’ teeth or offer food 2 h before oral sample collection. Saliva samples were collected using SalivaBio Infant’s Swab (SIS) (Salimetrics, Inc., Carlsbad, CA, USA). Plague samples were collected using a standard dental scaler. Oral samples were stored on ice after collection and transferred to the lab within 2 h for microbiological assays. The methods used to identify and quantify *Candida* spp. and *S. mutans* have been previously described [9,27]. Briefly, *S. mutans* was isolated using Mitis Salivarius with Bacitracin selective medium by incubating it at 37 °C for 48 h and identified by colony morphology [28]. BBL^TM^ CHROMagar^TM^ Candida (BD, Sparks, MD, USA) was used to isolate *Candida* spp. by incubating it at 37 °C for 48 h. Colonies of *S. mutans* and *Candida* spp. including *C. albicans*, *C. krusei*, and *C. glabrata,* were counted and recorded as colony forming units (CFU). Additionally, *C. albicans* and *S. mutans* were further identified using the colony polymerase chain reaction (PCR) method. 

### 2.3. Snack Data Collection

A questionnaire was administered at each visit to collect information on the child’s snack intake, including details on the amount and frequency of consumption of 15 common snacks and drinks, adapted from previous studies [29,30]. These snacks and drinks were categorized into a high or low cariogenic potential. Items, including chips, crackers, cookies, candy, soda or diet soda, dried fruit, ice cream, and fruit drink with sugar, were classified as high cariogenic potential and were summed to create a sweet index. Consumption of each snack/drink was assigned a score of “1”, while no consumption was assigned a score of “0”, according to previous studies [29,30]. The sweet index, representing intake of high-cariogenic snacks and drinks, ranged from 0 to 8. Items, including yogurt, dry cereal, fresh fruit, water, 100% juice, fruit drinks without sugar, and milk, were classified as low cariogenic potential and were summed to create a non-sweet index. The non-sweet index, indicating the intake of low-cariogenic snacks and drinks, ranged from 0 to 7. Sweet and non-sweet indices weighted by the amount and frequency of consumption were also assessed.

### 2.4. Covariates

Demographic information, oral hygiene practices, and feeding methods were obtained through questionnaires. Information on medications was extracted from medical records. Several variables were adjusted for in the analysis, including maternal education (high school or less vs. more than high school), infant sex, black race, white race, whether the father provided care to the infant (yes/no), tooth brushing practice (yes/no), exclusive breastfeeding at 12 months (0 = only solid food, 1 = breastfeeding exclusively; 2 = bottle feeding or both), history of antibiotics use until the time of visit (yes/no), and the number of erupted teeth. Plaque score and ECC were assessed at each visit by dentists in a dedicated examination room using standard dental examination equipment, materials and supplies [26]. Plaque score and ECC were also adjusted in the analysis. 

### 2.5. Statistical Analysis

The presence of oral cariogenic microorganisms, including *S. mutans* and *Candida,* was the primary outcome. We hypothesized that both sweet and non-sweet indices were associated with the carriage of cariogenic microorganisms, including *S. mutans* and *Candida*. Mixed-effects models were conducted to assess the associations between sweet and non-sweet indices of snack intake and oral microorganism carriage at 12, 18, and 24 months of age. Additionally, the potential modification effect of time was evaluated by including an interaction term between the time of visit and snack indices in the models. Logistic regression models were used to assess the associations between sweet and non-sweet indices and oral microorganism carriage at each study visit. The relative importance of 30 demographic, oral hygiene practice, and diet factors in relation to the carriage of oral cariogenic microorganisms in prediction was determined using the random forest method. All statistical analyses were performed using STATA 18.0 (College Station, TX, USA). 

Power analysis was conducted using G*Power 3.1 to determine adequate sample size for a cross-sectional logistic regression model. A mixed-effects model could potentially have higher power than correspondent cross-sectional logistic regression models due to its ability to account for within-subject correlations in a cohort study. Three effect sizes were assumed, corresponding to odds ratios of 1.2 (small effect size), 1.5 (medium effect size), and 3 (large effect size). With a significance level (*p*) set at 0.05, power at 80%, and an R-squared value of 0.2 between the predictor and other covariates, the sample sizes needed were calculated to be 482, 123, and 57, respectively, for the three effect sizes. According to this power analysis, our study had adequate power to identify a predictor with an odds ratio of 1.5 or more. 

## 3. Results

### 3.1. Characteristics of the Child Cohort

In this study, the majority of the children were African Americans (>53%), and more than half of the mothers had a high school education or less (>54%) (Table 1). Additionally, a significant proportion of the children were not exclusively breastfed at 12 months of age (>84%). The sweet and non-sweet indices of snack intake showed an increasing trend with age (*p* < 0.001). Analysis of children’s saliva and plaque samples revealed the presence of *S. mutans* and *Candida* in early life. The prevalence of *S. mutans* carriage in both saliva and plaques increased significantly with age (*p* < 0.001), rising from 21% or more at 12 months to 50% or more at 24 months. However, there was no significant age-related increase in *Candida* carriage in saliva or plaques (*p* > 0.17).

### 3.2. Association between Snack Intake and Cariogenic Microorganism Carriage

The associations between sweet and non-sweet indices and the carriage of *S. mutans* and *Candida* in saliva and plaques among children aged 12 to 24 months were analyzed using the mixed-effect models (Table 2). Higher scores on the non-sweet index were associated with increased odds of *S. mutans* carriage in plaques (OR = 1.67, 95%CI: 1.14, 2.46). The weighted sweet and non-sweet indices had similar results to the non-weighted indices, except that the weighted non-sweet index scores were positively associated with *Candida* carriage in plaques (OR = 1.01, 95%CI: 1.00, 1.02) (Appendix A). The modification effect of time on the relationship between snack intake and cariogenic microorganism carriage is presented in Table 3. Significant interactions between the sweet index and the time of visit were found in the associations with *Candida* carriage in saliva and plaques. Specifically, higher sweet index scores at 12 months were associated with an increased risk of *Candida* carriage in saliva and plaques (Figure 2A,B), whereas this association reversed at 24 months. An interaction was also observed between the non-sweet index and the time of visit (*p* = 0.04) in the association with *S. mutans* carriage in plaques but not in saliva (Figure 2C). The positive associations between non-sweet index scores and *S. mutans* carriage in plaques seemed to strengthen with age (Figure 2D). The weighted sweet and non-sweet indices had similar results to the non-weighted indices, except that the association between weighted non-sweet index scores and *S. mutans* carriage in plaques was not modified by the time of visit. But the association between the weighted sweet index scores and *S. mutans* carriage in plaques was modified by the time of visit, with a positive association more prominent in the earlier visit (Appendix A).

The cross-sectional associations between sweet and non-sweet indices and the carriage of *S. mutans* and *Candida* in saliva and plaques among children at each study visit are presented in Appendix A. Higher sweet index scores were associated with a decreased risk of *Candida* carriage in saliva at 24 months of age (OR = 0.73, 95%CI: 0.55, 0.98). Higher non-sweet index scores were associated with an increased risk of *S. mutans* carriage in plaques at 18 months of age (OR = 3.18, 95%CI: 1.44, 7.01). 

### 3.3. Rank of the Predictive Factors of Cariogenic Microorganism Carriage

The relative importance of 30 demographic, dental, and diet factors in relation to oral *S. mutans* and *Candida* carriage in prediction were determined using random forest. The sweet and non-sweet indices emerged as top-ranking factors for *S. mutans* carriage in both saliva and plaques between 12 and 24 months (Appendix A). Among individual snacks, cookie intake ranked among the top five predictive factors for *S. mutans* carriage in saliva at 12 months, while yogurt intake was among the top five predictive factors for *S. mutans* carriage in plaques at 12 and 24 months. Similarly, the sweet and non-sweet indices were identified as top-ranking factors for *Candida* carriage in both saliva and plaques between 12 and 24 months (Appendix A). Among individual snacks, cookie intake was among the top five predictive factors for *Candida* carriage in saliva and plaques at 18 months. 

## 4. Discussion

In this study, conducted on a birth cohort from low socioeconomic backgrounds, higher scores on the non-sweet index were linked to increased odds of *S. mutans* carriage in plaques, with this positive association potentially strengthening with age. On the other hand, higher sweet index scores at 12 months were linked to an increased risk of *Candida* carriage in saliva and plaques, with this association reversed at 24 months. Results from random forest indicated both the sweet and non-sweet indices as top-ranking factors in prediction for the carriage of *S. mutans* and *Candida*, with specific snacks like cookies and yogurt intake also featuring prominently. These findings underline the importance of both sweet and non-sweet snack consumption in their association with the presence of cariogenic oral microorganisms during early childhood, particularly among socioeconomically disadvantaged populations. 

Previous studies have presented mixed results regarding the association between sugar intake or sweetened food/beverages and the presence of *S. mutans* in saliva or plaques. Some studies have reported positive associations [19,20,22,25], while others found no association or low correlation [21,24]. In our cross-sectional analysis, a trend, albeit non-significant, of positive associations between sweet index scores and *S. mutans* carriage in saliva and plaques was observed at 12 and 18 months of age. This trend was also evident in the association between weighted sweet index scores and *S. mutans* carriage in saliva and plaques. Interestingly, several studies have also noted a positive relationship between eating or snacking frequency, including both sugary and non-sugary foods, and *S. mutans* carriage [20,23,25]. The study found a positive association between non-sweet index scores and *S. mutans* carriage in plaques, potentially strengthening with age. These results, along with previous research, suggests that snacks, regardless of sugar content, may contribute to *S. mutans* colonization. Additionally, certain snacks previously classified as low-cariogenic foods, such as dry cereal, may have a high sugar content [29]. The Environmental Working Group has reported that children’s cereals are more heavily loaded with added sugar compared to adult cereals [31]. These findings highlight the need for further investigation into the classification of the cariogenic potential of individual food and the specific mechanisms linking snack intake, particularly low- or non-sugary snacks, with the carriage of cariogenic microorganisms. 

This study has uncovered a novel connection between sweet index scores (both weighted and non-weighted) and oral *Candida* carriage. At 12 months of age, positive associations were found between sweet index scores and increased risk of *Candida* carriage in saliva and plaques. But at 24 months of age, negative associations were identified. To our knowledge, no prior studies have examined the relationship between dietary intake and oral *Candida* carriage in children. Nonetheless, in vitro studies have indicated that sucrose can promote *Candida* growth and the formation of inter-kingdom biofilms between *Candida* and *S. mutans* [32,33]. However, the observed variation in the association between the sweet index and *Candida* carriage with age in this study warrants further investigation to elucidate the underlying mechanisms.

The longitudinal and random forest analyses conducted in this study emphasize the potential importance of sweet and non-sweet indices in relation to the oral carriage of cariogenic microorganisms, *S. mutans* and *Candida*, in early childhood. However, several limitations should be considered when interpreting the results. Firstly, the study only assessed the intake of common snacks and drinks, and the main meals consumed by the children were not assessed. Future studies should incorporate more comprehensive dietary assessment, such as a diet diary or repeated 24 h dietary recalls, to provide a holistic understanding of dietary influences on the oral microbiome. Additionally, this study did not explore the association between snack intake and the quantity of *S. mutans* and *Candida* in the oral cavity, as a significant number of children in the cohort were free from *S. mutans* or *Candida* carriage. Future research could investigate the relationship between dietary intake and *S. mutans* or *Candida* growth among young children who test positive for *S. mutans* or *Candida* carriage. 

## 5. Conclusions

Findings from this study support the associations between snack intake and the presence of oral cariogenic microorganisms, *S. mutans* and *Candida*, in early childhood, underscoring the importance of considering dietary factors in oral health management for young children, particularly in underserved populations with limited access to dental care and healthy foods. Future research can expand on these findings by conducting more comprehensive dietary assessments and further elucidating the specific mechanisms by which different types of food and beverages influence the colonization of cariogenic microorganisms in order to inform targeted interventions. 

## Figures and Tables

**Figure 1 nutrients-16-01113-f001:**
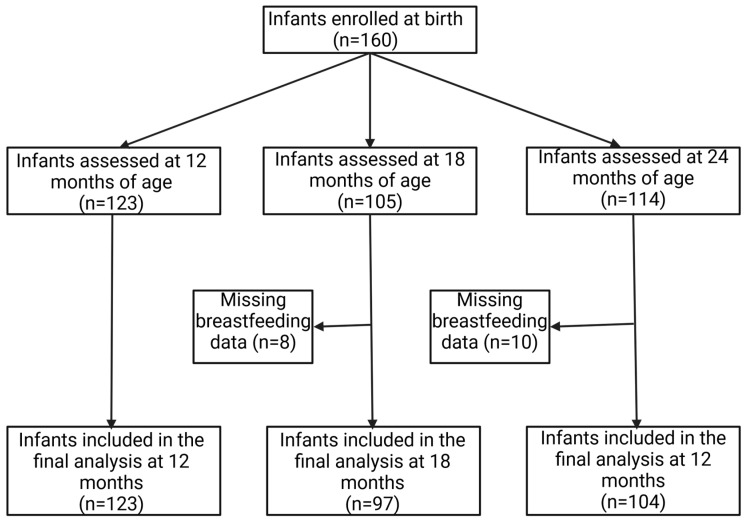
Participant flow diagram of the study.

**Figure 2 nutrients-16-01113-f002:**
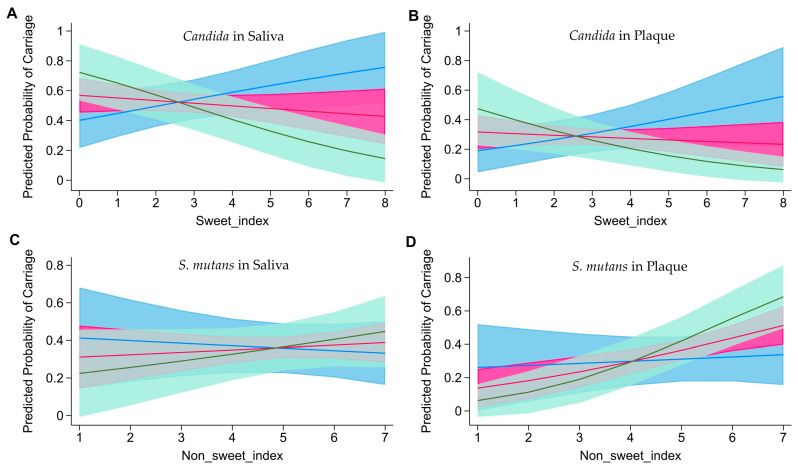
(**A**) The relationship between sweet index scores and predicted probability of *Candida* carriage in saliva at 12, 18, and 24 months. (**B**) The relationship between sweet index scores and predicted probability of *Candida* carriage in plaques at 12, 18, and 24 months. (**C**) The relationship between non-sweet index scores and predicted probability of *S. mutans* carriage in saliva at 12, 18, and 24 months. (**D**) The relationship between non-sweet index scores and predicted probability of *S. mutans* carriage in plaques at 12, 18, and 24 months. The blue line represents the relationship at 12 months and the blue area represents the confidence interval; the red line represents the relationship at 18 months and the red area represents the confidence interval; and the green line represents the relationship at 24 months and the green area represents the confidence interval.

**Table 1 nutrients-16-01113-t001:** Characteristics of the children (*n* = 123).

Characteristics	Percentage or Mean (SD)		
	12 Months ^a^ (*n* = 123)	18 Months ^a^ (*n* = 97)	24 Months ^a^ (*n* = 104)
Race__White	24.4%	23.7%	24.0%
__Black	54.5%	53.6%	54.8%
Female	51.2%	50.5%	51.9%
Maternal education (High school or less)	56.1%	54.6%	55.8%
Dad as care provider	43.1%	48.5%	45.2%
Exclusively breastfeeding at 12 months	15.5%	15.5%	13.5%
History of antibiotics use	12.2%	16.5%	18.3%
Tooth brushing	66.7%	92.8%	99.0%
Number of erupted teeth	5.8 (2.7)	13.2 (3.2)	17.1 (1.9)
Plaque score	0.24 (0.42)	0.49 (0.63)	0.63 (0.68)
ECC	3.3%	12.4%	25.0%
* **Predictors** *			
Snack__Sweet index	2.1 (1.5)	3.1 (1.7)	3.6 (1.7)
Snack__Non-sweet index	4.5 (1.4)	5.2 (0.9)	5.2 (1.1)
* **Outcomes** *			
*S. mutans* carriage__saliva	21.1%	41.2%	50.0%
*S. mutans* carriage__plaque	21.7%	36.1%	56.7%
*Candida* carriage__saliva	50.4%	54.6%	40.4%
*Candida* carriage__plaque	25.8%	23.7%	29.8%

Note. ^a^ Percentage or mean (SD).

**Table 2 nutrients-16-01113-t002:** Associations between snack intake and oral microorganism carriage from children aged 12 months to 24 months.

	Sweet Index			Non-Sweet Index		
	OR	95% CI	*p*	OR	95% CI	*p*
*S. mutans* carriage__saliva	1.05	0.85, 1.29	0.68	1.06	0.80, 1.42	0.68
*S. mutans* carriage__plaque	0.99	0.79, 1.25	0.95	**1.67**	**1.14, 2.46**	**0.01**
*Candida* carriage__saliva	0.91	0.75, 1.09	0.31	1.13	0.89, 1.45	0.32
*Candida* carriage__plaque	0.93	0.75, 1.16	0.54	0.81	0.61, 1.07	0.13

Note. Mixed-effects models were used to estimate the associations between sweet/non-sweet indices and oral microorganism carriage, respectively. Covariates adjusted in the models included maternal education, infant sex, infant race, dad as care provider, history of antibiotics use, tooth brushing, number of erupted teeth, exclusive breastfeeding at 12 months of age, plaque score, ECC, and time of visit. OR is odds ratio. CI is confidence interval. Bolded results are results that fulfill *p* < 0.05.

**Table 3 nutrients-16-01113-t003:** Interactions between sweet/non-sweet indices and time of visit and their associations with oral microorganisms.

	*S. mutans*				*Candida*			
	Saliva		Plaque		Saliva		Plaque	
	OR (95%CI)	*p*	OR (95%CI)	*p*	OR (95%CI)	*p*	OR (95%CI)	*p*
Sweet Index	1.70 (0.80, 3.64)	0.17	1.54 (0.66, 3.56)	0.26	2.69 (1.34, 5.43)	0.006	2.80 (1.29, 6.09)	0.01
Time	1.12 (0.93, 1.35)	0.22	1.16 (0.94, 1.42)	0.17	1.17 (0.99, 1.38)	0.06	1.17 (0.98, 1.40)	0.04
Sweet Index × Time	0.97 (0.94, 1.01)	0.19	0.98 (0.94, 1.02)	0.29	**0.94 (0.91, 0.98)**	**0.002**	**0.94 (0.90, 0.98)**	**0.004**
Non-sweet Index	0.64 (0.26, 1.60)	0.34	0.45 (0.13, 1.56)	0.21	1.37 (0.63, 2.95)	0.43	1.25 (0.53, 2.95)	0.62
Time	0.86 (0.64, 1.16)	0.34	0.74 (0.50, 1.09)	0.26	1.02 (0.79, 1.32)	0.86	1.11 (0.83, 1.47)	0.49
Non-sweet index × Time	1.03 (0.98, 1.08)	0.25	**1.08 (1.00, 1.16)**	**0.04**	0.99 (0.95, 1.03)	0.62	0.97 (0.93, 1.02)	0.30

Note. Mixed-effects models were used to estimate the interactions between sweet/non-sweet indices and time of visit, respectively. Covariates adjusted in the models included maternal education, infant sex, infant race, dad as care provider, history of antibiotics use, tooth brushing, number of erupted teeth, exclusive breastfeeding at 12 months of age, plaque score, and ECC. OR is odds ratio. CI is confidence interval. Bolded results are interaction results that fulfill *p* < 0.05.

## Data Availability

The original contributions presented in the study are included in the article/Appendix A, further inquiries can be directed to the corresponding author.

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
