# Peer review of "The Associations between Snack Intake and Cariogenic Oral Microorganism Colonization in Young Children of a Low Socioeconomic Status"

_nutrients, 2024, doi:10.3390/nu16081113_

Round 1

Reviewer 1 Report

Comments and Suggestions for Authors

The authors attempt to assess the association between snack intake and S Mutans/ Candida which is definitely an interesting question. Unfortunately, I am unsure of the validity of the associations found in this study for the following reasons (this  may be because of missing information in the M&M)

1. Oral sample collection: Did the research team ensure that no antimicrobials were used within 2 weeks of collection to ensure accurate reflection of the SM and candida load? Other confounders include the home oral hygiene regimen and the day of collection (e.g., use of fluoridated toothpaste, toothburshing on day of collection), perhaps authors should mentioned these confounders if they were accounted for but not mentioned in the  manuscript.

2. A diet diary would have been more accurate than just assessing the frequency of intake of 15 common snacks (as it was unclear whether all participants were exposed to these snacks and if so how common). It was unclear how the sweet/non-sweet indexes were derived? Has this scale been validated?

3. Plaque accumulation and presence of caries wasnt adjusted for and this would impact MS and potentially candida levels?

Reviewer 2 Report

Comments and Suggestions for Authors

Researcg gap: The research gap for this study is the unknown relationship between snack intake and cariogenic microorganisms like Candida and S. mutans in young children. It is a good question to be answered.

Aim: The study aim is to assess this association in a prospective underserved birth cohort. Data from children aged 12 to 24 months, including oral microbial assays and snack intake information, were analyzed. The aim is well defined.

Abstract. The abstract is well written. A conclusion is ideal before the last sentence reporting the significance of the study.

Introduction: Reporting the recent epidemiological studies on ECC status and Socioeconomic Status will strengthen the rationale for this studies.

Method: Define the type of this study and report what guideline they use for reporting of this research. Define the primary outcome measured, sample size calculation and hypothesis.

Results:

Table 1 and 2 can be formatted better for presentation. I suggest arranging the 3 diagram in Figure 2 (and also table 3  by two main role as S. m and Candida) in vertical for easy reading.

Discussion: Concise discussion.

Conclusion: can be shorten to address the aim of the study.

Comments on the Quality of English Language

Easy to follow.
